# MicroRNAs in Age-Related Proteostasis and Stress Responses

**DOI:** 10.3390/ncrna9020026

**Published:** 2023-04-13

**Authors:** Latika Matai, Frank J. Slack

**Affiliations:** Department of Pathology, Beth Israel Deaconess Medical Center, Harvard Medical School, Boston, MA 02115, USA

**Keywords:** aging, miRNA, miR, proteostasis, stress response, heat-shock, HSPs, UPR, autophagy, longevity, lifespan, health-span

## Abstract

Aging is associated with the accumulation of damaged and misfolded proteins through a decline in the protein homeostasis (proteostasis) machinery, leading to various age-associated protein misfolding diseases such as Huntington’s or Parkinson’s. The efficiency of cellular stress response pathways also weakens with age, further contributing to the failure to maintain proteostasis. MicroRNAs (miRNAs or miRs) are a class of small, non-coding RNAs (ncRNAs) that bind target messenger RNAs at their 3′UTR, resulting in the post-transcriptional repression of gene expression. From the discovery of aging roles for *lin-4* in *C. elegans,* the role of numerous miRNAs in controlling the aging process has been uncovered in different organisms. Recent studies have also shown that miRNAs regulate different components of proteostasis machinery as well as cellular response pathways to proteotoxic stress, some of which are very important during aging or in age-related pathologies. Here, we present a review of these findings, highlighting the role of individual miRNAs in age-associated protein folding and degradation across different organisms. We also broadly summarize the relationships between miRNAs and organelle-specific stress response pathways during aging and in various age-associated diseases.

## 1. Introduction

Aging comes with the major risk factor for various age-associated diseases including neurodegenerative diseases such as Alzheimer’s or Parkinson’s. Loss of protein homeostasis (or proteostasis) is the primary hallmark of many of these dementias, leading to the accumulation of misfolded protein aggregates with age [1]. Proteostasis is the process of regulating proteins within the cell to ensure the health of the cellular proteome and organism. A complex proteostasis network comprising the translational apparatus, molecular chaperones or foldases, and the proteolytic machinery ensures the maintenance of the cellular proteome in a healthy organism [2]. Additionally, cells employ different stress response machineries such as the heat stress response, unfolded protein response, or oxidative stress response that detect the rise in misfolded proteins and ensure the sustenance of proteome integrity [3,4,5,6]. The efficiency of the proteostasis network and the stress response pathways gradually declines with age, thus affecting proteome maintenance [7,8].

MicroRNAs (miRNAs) are short ncRNAs that repress the translation or induce mRNA degradation of target transcripts through sequence-specific binding to the target mRNA 3′UTR [9]. The biogenesis of miRNAs begins with their transcription in the nucleus under the control of RNA polymerase II to a 70–110 nucleotide (nt) long primary miRNA (pri-miRNA) [10]. This is further processed by DROSHA, an RNAse III enzyme along with its co-factor DGCR8, to generate a precursor miRNA (pre-miRNA) [11]. The pre-miRNA is exported to the cytoplasm with the help of exportin-5, where it is further cleaved by another RNAse III enzyme, DICER, into a miRNA duplex [12,13]. Strand separation and unwinding later forms a 22 nt long mature miRNA strand, which binds its target mRNA along with the Argonaute proteins in the RNA-induced silencing complex (RISC) [13].

The role of miRNAs in controlling aging processes has been recently demonstrated with the discovery of miRNAs that regulate lifespan in the nematode *Caenorhabditis elegans* [14,15]. To date, numerous miRNAs have been found to be differentially regulated with age, and many of these have emerged as regulators of aging at both the tissue and organismal level [16]. Recent studies have also revealed the role of miRNAs in regulating proteome integrity, either by directly targeting components of the protein folding or degradation machinery, or indirectly by targeting components of various stress response pathways, as discussed here.

In this review, we will first introduce miRNAs that regulate longevity in *C. elegans* and *Drosophila*, together with the recent advances in their roles in mammalian aging. Next, we review the role of miRNAs in maintaining proteome homeostasis/proteostasis at the organismal or cellular level with an emphasis on the miRNA-mediated regulation of protein folding and degradation. Finally, we discuss studies that identify the relationships between miRNAs and cellular stress defense mechanisms with age and discuss interesting prospects in unveiling the roles of miRNAs in age-associated collapse in stress responses, proteostasis, and lifespan.

## 2. miRNAs in Aging

Aging is a complex, physiological process characterized by the progressive loss of function, degradation, and decline in the repair capacity of animal tissues and organ systems. Findings from model organisms have revealed that aging can be manipulated by genetic, epigenetic, or environmental factors. miRNAs are being increasingly recognized as regulators of longevity and aging processes [15]. They also serve as potential aging biomarkers [17,18]. Since the discovery of the first miRNA, *lin-4* in *C. elegans,* over 1900 miRNAs have been annotated and published in miRbase, regulating approximately 60% of the human transcripts, although very few have been studied for their role in aging processes [19,20].

miRNAs are critical regulators of longevity and regulate lifespan through well-defined aging pathways such as insulin/insulin-like growth factor signaling (IIS), dietary restriction (DR), target of rapamycin (TOR), DNA damage responses, sirtuins (histone deacetylases), and reactive oxygen species signaling [15]. Most knowledge about miRNAs as the modulators of lifespan have come from studies on invertebrate model organisms such as *C. elegans* and *Drosophila*. *lin-4* was the first miRNA to be reported to influence lifespan in worms by regulating the expression of its target gene *lin-14* [14]. Since then, the role of several miRNAs has been characterized in modulating longevity and their levels are also regulated with age [21,22]. The loss of function (*lof*) of *mir-71, mir-238*, *mir-246*, and the *mir-229,64,65,66* cluster is known to reduce lifespan, whereas *mir-239* and *mir-228 lof* promote lifespan [21,22,23,24,25]. miR-71 is required for lifespan extension induced by either germline loss or dietary restriction, while miR-228 and miR-229,64,65,66 are required for DR-mediated longevity [23,24,25]. In addition to lifespan modulation, these miRNAs are known to affect heat and oxidative stress resistance in these nematodes, suggesting that they play critical roles in maintaining homeostasis at cellular and organismal levels. Further, miRNAs also act as biomarkers to predict longevity in worms. For instance, the levels of miR-71, miR-238, and miR-246 correlated positively with lifespan in wild-type individuals, while levels of miR-239 corelated negatively with lifespan [17]. Interestingly, even though several studies indicate the indispensable role of miR-71 in adult lifespan, it is not conserved in higher organisms, unlike miR-34. Recent studies indicate that the loss of miR-34 significantly extends the wild-type lifespan in *C. elegans* by increasing the suppression of the autophagy genes *atg-4.1, bec-1*, and *atg-9* and enhancing stress survival [26]. Lastly, loss of Argonaute-like gene-1 (*alg-1*), which is required for miRNA processing and gene silencing, is also known to reduce lifespan in *C. elegans* and, like individual miRNAs, is linked to stress resistance and proteostasis [27,28].

In *Drosophila*, several miRNAs have been characterized to both positively and negatively influence longevity and this effect is generally sex specific. For example, *mir-125* or *let-7* loss of function reduces lifespan in males, while their overexpression decreases lifespan in females [29,30]. Additionally, there are other miRNAs (*mir-282, mir-277*) for which their loss of function results in reduced lifespans while their overexpression also induces larval lethality or shortened lifespans [31,32]. A few conserved or homologous miRNAs display similar or opposite effects on longevity compared to their orthologs in other species such as *C. elegans.* For instance, the loss of conserved miRNA *mir-34* reduces lifespan and promotes age-associated neurodegeneration in *Drosophila* compared to its anti-longevity role in *C. elegans* [33]. miR-1000 is also another neuroprotective miRNA with a positive role in lifespan regulation [34]. In contrast, the loss of *mir-305*, a fly ortholog of *mir-239* in *C. elegans,* promotes lifespan extension, while its overexpression reduces lifespan and aggravates age-related decline in movement and proteostasis [35].

Lifespan-modulating miRNAs have been less extensively studied in mammalian species and miR-17 is the first miRNA reported to extend lifespan when ubiquitously expressed in mice [36]. It directly targets insulin receptor substrate (*Irs1)* and adenylate cyclase 5 (*Adcy5*), which upregulates the expression of *Mkp7*, a phosphatase that dephosphorylates and inhibits mTOR. Silencing *Irs1* or *Adcy5* also promoted autophagy and inhibited senescence and apoptosis. In addition, the brain-specific knockdown (KD) of miR-29 affects the mouse lifespan in a sex-specific manner where brain miR-29 KD extends male longevity, but shortens female longevity. Interestingly, miR-29 brain KD had the opposite effect on male and female fertility and was the first miRNA studied to have implications on the lifetime trade-off between two fitness components: aging and reproduction [37]. Several studies also highlight potential biomarkers for mammalian lifespan. Three miRNAs, namely, miR-203-3p, miR-664-3p, and miR-708-5p, have recently been discovered as potential biomarkers for median lifespan in mice and potentially regulate key longevity pathways including mammalian target of rapamycin (mTOR), FOXO, and mitogen-activated protein kinase (MAPK) [38]. In humans, a study measured the miRNA profiles of serum samples from groups that have longer lifespans (76–92) in comparison with a group with shorter lifespans (58–75) and found 24 significantly upregulated and 73 downregulated candidates. Six of these miRNAs (miR-211-5p, 374a-5p, 340-3p, 376c-3p, 5095, 1225-3p) have valid targets encoding the aging-associated genes *PARP1, IGF1R,* and *IGF2R* and therefore can potentially act as aging biomarkers [39]. The miRNAs that regulate lifespan in different species are summarized in Figure 1. It will be interesting to see whether other longevity-promoting miRNAs will be discovered in mammals regulating pathways relevant to human longevity.

The miRNAs in mammals are differentially expressed across multiple tissues and organs with age and appear to play both beneficial and detrimental roles in their aging. These topics have been comprehensively reviewed previously over several different articles; however, we cite a few examples below [40,41,42]. Several studies have indicated that the miRNA profile of the brain changes with age and some have been demonstrated to regulate brain aging and cognitive decline [42,43,44,45,46]. For example, miR-34c is upregulated during normal as well as pathological aging in mice and has been shown to target sirtuin1 (*Sirt1*) to modulate cognitive decline [47]. Another family member, miR-34a, is also upregulated with age and is known to target *Sirt1* and the anti-apoptotic protein Bcl-2 [48]. The increased aggregation of beta amyloid (Aβ) in the brain is a hallmark of brain aging and Alzheimer’s disease (AD) [49]. It is caused by the abnormal processing of amyloid precursor protein (APP) by the enzyme BACE1 [50]. Several miRNAs known to target *Bace1* are downregulated with age [51]. The overexpression of one of these, miR-186, is known to suppress *Bace1* and, hence, Aβ formation, suggesting that its downregulation with age results in age-related pathologies in the brain [52].

Several miRNAs have also been linked to muscle aging or sarcopenia. The miRNA profiling of the skeletal muscle of mice, primates, porcine, and humans reveled several miRNAs to be dysregulated with age [53,54,55,56,57,58,59,60]. A subset of these, including miR-434–3p and miR-431, are downregulated in aged myoblasts and play a protective role in age-associated apoptosis, while others such as miR-29 have been shown to increase with age and contribute to age-induced muscle loss [60,61,62,63].

The progressive deregulation of bone remodeling as well as bone loss is a consequence of normal aging in humans, and miRNAs are being increasingly recognized as playing a role in this complex process [64,65]. For instance, miR-183p is known to be highly expressed in bone marrow-derived extracellular vesicles and has been shown to inhibit bone formation [66]. It also induces osteoclast differentiation in bone marrow-derived macrophages by targeting heme oxygenase 1 (*Ho1)* [67]. Levels of another miRNA, miR-214 increases with age and is known to promote osteoclastogenesis by targeting *Pten* and the subsequent activation of the PI3K/AKT pathway [68]. In contrast, miR-21 plays a role in regulating osteocyte death by regulating the expression of the pro-apoptotic gene *Pten* [69]. With age, miR-21 was found to be suppressed by TNFα and downregulated FGF and ERK-MAPK signaling, thus impairing bone formation [70].

Levels of miRNAs such as miR-21, miR-22, and the miR-17-92 cluster are altered in normal cardiac aging and age-related cardiac pathologies where their roles have been previously described. Among these, levels of miR-21 and miR-22 have been found to be increased in aged cardiomyocytes and promote cardiac fibrosis and myofibroblast formation post myocardial infarction (MI) [71,72,73]. In contrast, levels of the miR-17–92 cluster, which consists of six individually transcribed miRNAs (miR-17, miR-18a, miR-19a, miR-19b, miR-20a, and miR-92a-1) are downregulated with age in heart failure-prone mice [74]. Among these, miR-17 is known to suppresses senescence and apoptosis by targeting proapoptotic protein PAR4 [75]. Overall, miRNAs are differentially regulated with age in different tissues and have also emerged as important regulators of aging at both the tissue and organismal level.

## 3. miRNA in Proteostasis Maintenance

Proteostasis/protein homoeostasis refers to the process of maintaining proteins in the desired concentration, conformation, and subcellular location by an extensive network of complex and integrated pathways that control the biogenesis, folding, and degradation of proteins [2]. Loss of proteostasis is one of the prime hallmarks of aging [8]. Misfolded and damaged proteins can accumulate as a result of the decline in proteostasis machinery with age, leading to the onset of age-associated protein misfolding diseases such as Huntington’s, Alzheimer’s, and Parkinson’s. Over the years, several studies have indicated the role of miRNAs in regulating components of the proteostasis network that work at various steps such as translation, folding, and degradation. Among these, the function of miRNAs in regulating protein translation is well established and has been comprehensively reviewed in several articles [76,77,78]. Interestingly, age is associated with significant alterations in the translational apparatus and its fidelity, as well as the rate of translation [79,80,81,82]. Multiple aging pathways such as the insulin–insulin growth factor 1 (IIS) pathway and TOR pathway seem to modulate different components of translational machinery and are themselves targets of several miRNAs [83,84]. Further, miRNAs also play a role in translational reprogramming, a phenomenon that is age-associated and comprises both the translation suppression and selective translations of several mRNAs [79,85,86,87,88]. However, the link between the three (miRNA, translational reprogramming, and aging) is still unclear.

miRNAs also regulate components of the protein folding network, particularly chaperones. Molecular chaperones are the class of proteins known to assist the folding of other proteins in cells and are sometimes referred as heat shock proteins or HSPs. They are present in every cellular compartment and are classified into five major classes based on their observed molecular weights: HSP60, HSP70, HSP90, HSP104, and the small HSPs [89]. There are numerous reports on the interaction between miRNAs and HSP40 chaperones, a class of small HSPs (also known as *DnaJ*-proteins), in age related pathologies. For instance, DNAJB1 is involved in the clearance of mutant polyglutamine (polyQ) protein Ataxin-3 aggregates and is suppressed in spinocerebellar Ataxia Type 3 (SCA3). miR-370 and miR-543, which are both upregulated in SCA3, are known to directly target DNAJB1 [90]. Another J protein, DNAJC3, is a target of the miR-200 family, and its miRNA-mediated suppression was demonstrated to have a role in pancreatic beta cell loss and obesity [91]. Other miRNAs such as miR-425-5p are detrimental for protein aggregation and chaperone expression in age-associated Alzheimer’s disease [92]. miR-425 is found upregulated in AD patients and targets heat shock protein B8 (HSPB8) and promotes tau phosphorylation in HEK293/tau cells [92]. The role of miR-425 in age-associated neuropathologies including Alzheimer’s is debatable. Studies have indicated that the loss of miR-425 promotes an amyloid plaque environment and neuronal loss in an AD mouse model and promotes the dopaminergic neurodegeneration of Parkinson’s mouse models; however, the role of foldases is still unexplored in these studies [93,94]. While the list of validated regulatory interactions of J-proteins with miRNAs is quite limited, the literature suggests there are a significant number of interactions awaiting validation. To this end, Budrass et al. studied the potential intersection of miRNA regulatory networks with the J-protein chaperone network using TargetScan, which revealed a considerable number of predicted miRNAs targeting J-proteins [95]. miRNAs also modulate the expression of heat shock proteins indirectly with age. A recent study in *Drosophila* pointed to the role of miR-34 in age-associated neurodegeneration, where the loss of miR-34 accelerated aging and protein aggregation. Notably, miR-34 targets *Pcl* and *Su(z)12*, two components of polycomb repressive complex 2, (PRC2), which is an H3K27me3 methyltransferase. H3K27me3 methyltransferase functions to trimethylate histone 3, which generally downregulates the expression of associated genes via the formation of heterochronic regions. Thus, the upregulation of PRC2 complex suppresses the expression of heat shock protein with age. Thus, miR-34 regulates the age-associated downregulation of HSPs and alleviates protein aggregation [96]. Furthermore, miRNAs also regulate chaperone-mediated autophagy (CMA) in age-associated neurogenerative diseases such as Parkinson’s disease (PD). Many cytosolic proteins are targeted for degradation in lysosomes by chaperone-mediated autophagy (CMA) where a peptide sequence motif is recognized by a heat shock cognate protein (HSC70) and internalized into lysosome by lysosomal-associated membrane protein 2A (LAMP2A) [97]. In PD, both HSC70 and LAMP2A are under the regulation of several miRNAs, some of which are also upregulated during the disease [98]. In particular, an increased expression of miR-106a, miR-26b, and miR-301b deregulates HSC70-mediated autophagy, and aggravates alpha-synuclein pathology [98]. Overall, miRNAs, in addition to their role in translation regulation, can regulate protein folding and serve as potential therapeutic targets to treat age-associated protein misfolding diseases (Figure 2).

Several recent reports have indicated an additional role of miRNAs in proteostasis, i.e., by regulating protein degradation. In *C. elegans,* the Argonaute proteins ALG-1 and ALG-2, which are known to affect nematode lifespan, positively regulate the proteasomal degradation within the cytosol- and ER-associated degradation (ERAD). The accumulation of UbV-GFP (a ubiquitin fusion degradation substrate, destined for degradation by 26S proteasome upon polyubiquitylation) is observed in animals undergoing knockdown of *alg-1* and *alg*-2. The accumulation of an ERAD substrate, CPL-1:YFP, is also observed in animals fed with *alg-2* RNAi, but not *alg-1* RNAi [28]. Another gerontomiR (miRNAs that regulate lifespan), miR-71, also promotes ubiquitin-dependent protein turnover to maintain proteostasis and longevity in nematodes. *mir-71(n4115)* mutants showed a substantial increase in both UbV-GFP and CPL-1*-YFP levels in intestine compared to wild-type animals. They also showed that miR-71 directly inhibits the toll/interlukin-1 receptor domain protein TIR-1 in AWC olfactory neurons and the *mir-71/tir-1* regulatory axis, which is important for the effect of food perception proteosome degradation and proteostasis maintenance [99].

Lastly, miRNAs also maintain proteostasis within ER by influencing the processes that regulate ER Ca^2+^ handling and storage in response to frequently changing intracellular and environmental conditions. Since ER Ca^2+^ concentration can directly influence the activity of ER-resident chaperones and stress response pathways, miRNAs do play an indirect role in maintaining ER proteostasis, as reviewed in [100]. Briefly, several miRNAs regulate the Ca^2+^ uptake and release from ER by targeting SERCA, I3PR, and RYR channels [100]. A few Ca^2+^ dependent ER chaperones are also miRNA targets; for example, calreticulin (CALR) contains miR-455-binding sites and is upregulated by ER stress, in part by the downregulation of miR-455 [101]. Another ER protein, protein disulfide isomerase 6 (PDIA6), is regulated by miR-322 and is upregulated during ER stress by the consecutive downregulation of miRNA, helping to restore ER calcium homeostasis [102]. The most abundant ER chaperone, BiP (heat shock 70 kDa protein 5 (HSPA5)), is regulated by the cooperative action of miR-30, miR-181, and miR-199-5p in several cancers when calcium homeostasis is disturbed. Thus, these miRNAs are therapeutic targets in order to prevent ER stress-dependent damage in these tumors [103].

## 4. miRNAs and Stress Responses

Organelle-specific stress responses play a central role in the proteostasis maintenance within the organelle as well as in the cytoplasm. These stress response machineries sense the accumulation of misfolded/damaged proteins or the depletion of foldases/degradation components through various sensors located in the cytoplasm or the organelle membrane. This initiates a cascade of events leading to either the decreased translation of proteins in order to minimize the load onto the proteostasis machinery, or the increased transcription of chaperones and proteasomal components in order to better assist the misfolded proteins [1,104]. Other roles for miRNAs have recently been identified as targeting different components of stress response pathways and are also differentially regulated during the activation of these cascades. In this section, we will review the role and regulation of miRNAs in four major cellular stress response pathways with age, namely, cytosolic heat shock response (HSR), endoplasmic reticulum unfolded protein response (UPR^ER^), oxidative stress response, and autophagy. The efficiency of these machineries is known to decline with age in different organisms, leading to a drop in proteostasis [7,105,106]. This section will highlight how miRNAs can play a significant role in this age-associated decline—a phenomenon that is currently understated. Additionally, it emphasizes certain miRNAs (such as miR-34) that target the components of several stress response pathways and are therefore the most well studied miRNAs for their potential role in proteostasis and health span regulation.

### 4.1. Heat Shock Response

The heat shock response (HSR) is the cellular protective mechanism that is induced by several stressors such as elevated temperatures and oxidative or chemical insults, which disrupt protein folding homeostasis [107,108,109]. Central to the activation of the heat shock response is the activation of the heat shock factor (HSF) transcription factor, which induces the expression of several heat shock proteins (HSPs). HSPs are molecular chaperones that help to restore protein folding or prevent further misfolding, ensuring proteostasis maintenance [110,111]. Heat stress also inhibits DNA synthesis, transcription, post-transcriptional processing, and translation, potentially leading to cell cycle arrest and subsequent protein degradation. The elevation of heat shock proteins is critical for the survival of eukaryotic cells at elevated temperatures. A negative feedback loop exists between HSF and HSPs during acute stress, and once the effect of the stress has been dealt with, the HSP levels are reduced to basal levels [112].

Post-transcriptional mechanisms can potentially regulate the levels of protein coding mRNAs during HSR [113,114,115]. The relationship between miRNAs and HSR is bidirectional and can be best understood with examples from aging studies performed in *C. elegans.* In *C. elegans,* several miRNAs have been found to be up- or downregulated during heat stress [116,117]. An in-depth phenotypic analysis of miRNA deletion mutants revealed four miRNAs—miR-71, miR-239, miR-80, and miR-229,64,65,66—to play critical roles in survival under heat stress [117]. Among these, miR-71 and miR-229-66 cluster promote longevity and stress resistance in worms, while miR-239 antagonizes them [22,23,25]. Further, the expression of these miRNAs is also regulated with age, and they target genes in the insulin signaling pathway (IIS) to regulate lifespan [21,22].

The transcriptional control of HSR is mediated by heat shock transcription factor 1 (HSF1) [118,119], which binds to the promoter of protein coding genes induced by heat shock. While the transcriptional response of protein coding genes under HSF-1 has been comprehensively analyzed in a variety of organisms, the regulation of ncRNAs has not been systematically examined. Brunquell et al. identified miRNAs that were differentially regulated in the presence or absence of HSF-1 with or without heat stress in *C. elegans* [116]. The study identified six miRNAs, miR-784, miR-231, miR-86, miR-53, miR-47, and miR-34, that are upregulated by HSF-1 under heat stress. Integrated miRNA/mRNA target prediction analyses suggested that HSF-1 controls the processes of development, metabolism, and longevity through the regulation of miRNA expression. Further, HSF-1-regulated miRNAs also provide a potential link between different stress response pathways and lifespan. Among the miRNAs upregulated by HSF-1, miR-34 is a highly conserved miRNA, with orthologs in *Drosophila,* mouse, and human. In *C. elegans*, the levels of miR-34 provide a robust response to environmental stress and are regulated by the insulin signaling pathway via a negative feedback loop between miRNA and DAF-16/FOXO [120]. miR-34 is also highly expressed during aging and modulates lifespan by regulating the autophagic flux [26]. In contrast, the loss of miR-34 accelerates aging and brain degeneration in *Drosophila* [33]. The potential link between other upregulated miRNAs and aging and/or stress response is yet to be determined. The miRNAs that are downregulated upon heat shock are miR-48 and miR-228. miR-228 is upregulated in aging nematodes, and *mir-228* deletion has been shown to increase longevity and heat stress resistance [24]. *mir-48* belongs to the *let-7* miRNA family that controls developmental timing in *C. elegans* [121]. Thus, HSF-1 might normally suppress miR-228 and miR-48 expression during HS to promote longevity and stress resistance, and to control developmental events during stress conditions.

A more recent study revealed that HS induces a two-fold change in approximately 5% of miRNAs and identified additional differentially regulated miRNAs. Among these, miR-4936, which is barely detectable under control temperature conditions, showed the most dramatic upregulation during HS [122]. Although this was largely consistent with the studies of miRNAs that are differentially regulated during heat stress, there were several differences. For instance, Schreiner et al. noticed that a significant downregulation miR-246 is detected upon stress, which is contrary to its previously reported role in promoting heat stress resistance. Another counter-indicative finding was that levels of miR-239 were significantly upregulated during stress and *mir-239* is transcriptionally induced by HSF-1 [122]. However, according to a previous study, *mir-239* mutants demonstrate enhanced survival at higher temperatures [22]. Thus, a detailed investigation of the molecular mechanism underlining the role of these miRNAs under heat stress is required. Additionally, most of the miRNAs that were found differentially regulated under heat stress or by HSF-1 (Table 1) have not yet been studied to ascertain their biological functions. Thus, further studies identifying the bona fide targets of these miRNAs and the downstream molecular mechanism will be necessary to understand their role and regulation during heat stress response.

Recovery from heat shock also requires the miRNA pathway. The depletion of miRNA Argonaute ALG-1 disables the suppression of HSP-70 levels after heat shock, thus affecting post-stress survival *in C. elegans*. Additionally, evidence suggests that the regulation of HSP-70 is driven by miR-85, the target sites of which are present in *hsp-70* mRNA’s 3′UTR. The downregulation of *hsp-70* by miR-85 promotes survival, and animals lacking this miRNA exhibit reduced viability post heat shock [123]. The study points to a previously unknown role of the miRNAs, where the downregulation of heat shock protein levels is necessary for post-stress survival in worms.

A study in *Drosophila* investigated the crosstalk between HSR and miRNA machinery across different fly strains. Compared to the control’s unstressed conditions, the miRNA profiles formed a uniform pattern of differential regulation in different strains under heat stress. Moreover, the study observed a general downregulation of precursor miRNA (pri-miRNA) transcripts as well as core miRNA pathway genes; however, the levels of mature miRNAs were found to be upregulated [124]. This suggests that the regulation of miRNA expression occurs at both the transcriptional and post-transcriptional level.

Another study conducted in the dermal fibroblasts showed the differential expression of 123 miRNAs following a hyperthermia-induced cellular stress response [125]. Target prediction analyses suggested that several HSPs and AGO2 (the core protein required for miRNA-dependent gene silencing) were putative targets for these miRNAs [125]. Another study identified differentially expressed miRNAs and their targets in injured heart, liver, kidney, and lung during a heat stress and recovery period in rats [126]. miR-21 was identified as the most differentially expressed miRNA in injured cardiomyocytes and its levels were also modulated in lung and kidney cells. Manipulating the levels of miR-21 in rat-derived myoblast (H9C2) cells by transfection with a miR-21 inhibitor significantly increased apoptosis in these cells following heat stress [126].
ncrna-09-00026-t001_Table 1Table 1miRNAs regulated during heat stress in *C. elegans.*
^#^ indicates miRNA found in both studies.miRNAs Upregulated during HSStudymiRNAs Downregulated during HSStudymiR-784, miR-355, miR-1829c, miR-62, miR-794, miR-46, miR-5592, miR-231, miR-65, miR-86, miR-84, miR-232, miR-63, miR-2212, miR-229miR-66, lin-4, miR-52, miR-53, miR-237, miR-1022, miR- 4816, miR-239b ^#^, miR-239a ^#^, miR-1830, miR-1820, miR-230 ^#^Brunquell et al., 2017 [116] *let-7*, miR-57, miR-51miR-64, miR-61, miR-75, miR-252, miR-83, miR-56, miR-36, miR-58, miR-795, miR-82, miR-2214, miR-55, miR-90, miR-235, miR-4926, miR-45, miR-73, miR-74, miR-77, miR-35, miR-4813, miR-87, miR-44, miR-238miR-250, miR-42miR-40miR-54, miR-39miR-41, miR-37, miR-246 ^#^, miR-67, miR-47, miR-34 ^#^, miR-355, miR-71miR-790 ^#,^ miR-79^,^ miR-38Brunquell et al., 2017 [116]miR-4936, miR-247, miR-235, miR-797, miR-788^#^ indicates miRNA found in both studies Schereiner WP.et al, 2019 [122]miR-1817, miR-85, miR-5592, miR-240, miR-359, miR-50, miR-59, miR-358Schereiner WP.et al, 2019 [122]

### 4.2. ER Stress Response

Different cellular and environmental challenges, such as nutrient limitation, oxygen deprivation, or oxidative stress, can affect the protein folding capacity of the endoplasmic reticulum (ER). The accumulation of misfolded proteins within the ER triggers an adaptive signaling pathway coined the unfolded protein response (UPR) or ER stress response. This adaptive stress response prevents an excess of misfolded or aggregation-prone proteins in the ER lumen and is important for maintaining proteostasis. The UPR^ER^ consists of three different branches, each represented by sensor proteins: inositol-requiring enzyme 1a (IRE1a), activating transcription factor 6 (ATF6), and protein kinase R such as endoplasmic reticulum kinase (PERK) [127,128]. Upon ER stress, BiP, one of the most abundant ER chaperones, dissociates from these three sensors, leading to their activation through either oligomerization or export [129]. Upon activation, PERK phosphorylates eukaryotic initiation factor 2 alpha (eIF2α), which results in reduced translation initiation at many mRNA transcripts, although it increases the translation of activating transcription factor 4 (ATF4) [130]. IRE1a oligomerizes in response to the increased abundance of misfolded proteins in the ER lumen, which leads to trans-autophosphorylation through its cytoplasmic kinase domain and activates its site-specific endonucleolytic activity [131]. This promotes the unconventional splicing of the mRNA-encoding X-box binding protein (XBP1) [132]. Later, the translated product of spliced XBP1 functions as a transcription factor promoting the expression of UPR^ER^ genes [133]. ATF6 undergoes proteolytic processing that yields an active transcription factor (ATF6(N)), which upregulates the expression of various ER-resident quality control proteins, including chaperones and ER-associated degradation (ERAD) components [134,135,136].

The involvement of both miRNAs and UPR^ER^ has been individually reported in regulating aging; however, the number of studies examining their relationship in the context of aging are very limited. Thus, only a few examples that demonstrate the effect of miRNAs on UPR or vice versa can be used to evaluate their potential interactions during aging. For instance, in *C. elegans, mir-71* mutant animals, which are shorter lived than wild-type animals, are also sensitive to ER stress induced by tunicamycin (TM), a drug that blocks the N-linked glycosylation and subsequent folding of ER proteins [99]. As miR-71 regulates ubiquitin-dependent protein turnover, it can be hypothesized that *mir-71* mutants might be experiencing a misfolded protein load that sensitizes them toward ER stress. Alternatively, miR-71 could also be directly regulating key housekeeping genes within the UPR branches, a topic that requires further investigation.

Argonaute proteins, which influence lifespan across different species and are normally localized in the cytoplasm under basal conditions, are found in stress granules near ER along with miRNAs [137,138,139]. Stress granules are cytoplasmic, membrane-less ribonucleoprotein assemblies that contain translationally stalled mRNAs, ribosomal components, translation initiation factors, and RNA-binding proteins (RBPs) [140,141]. The phosphorylation of eiF2α by PERK during ER stress induces translational arrest that enables the release of mRNAs from polysomes and the nucleation of stress granules. Further, miRNAs are required for AGO localization to stress granules, suggesting a role of miRNAs during ER stress [139]. Aging, which is accompanied by a decrease in protein quality control systems, can lead to the formation of aberrant stress granules [142]. Thus, the relationship between UPR and Argonaute proteins in age-associated stress granule formation should be further investigated.

Several studies demonstrate the existence of the ER stress-dependent regulation of miRNA expression. The activation of IRE1 under stress has been shown to target several mRNAs for degradation in an XBP1-independent manner, a mechanism known as regulated IRE1-dependent decay (RIDD) [143]. RIDD potentially relieves the load on the ER by mediating the degradation of ER-targeted mRNAs. ER stress-mediated RIDD has been demonstrated for many miRNAs including miR-34, miR-200, miR-17, miR-96, and miR-125b [144,145]. Another example that might indicate a close association with aging is the RIDD of miR-34. This is because miR-34, in addition to its role in lifespan regulation across different organisms, is also known to regulate the chaperone system and proteostasis, and plays a vital role in brain aging in *Drosophila* [26,33,96,146]. It would be beneficial to further investigate miR-34 in aging and its regulation via RIDD.

In turn, miRNAs have also been shown to regulate the ER stress response and carry an understated relationship with age. For example, ER stress activates the transcription factor XBP1 via IRE1-mediated non-canonical splicing. This later induces the expression of genes involved in protein folding or ER-associated degradation (ERAD). The expression of XBP1 declines with age and its overexpression has been demonstrated to promote lifespan as well as protection against pathological tau [147,148,149]. Byrd et al. showed that miRNA-30c-2* targets a single site in the 3′UTR of *XBP1* and thus influences the survival of cells experiencing ER stress [150]. The miRNA is upregulated during UPR^ER^ in a PERK-dependent manner. Further studies will demonstrate the role of this miRNA in regulating the expression of XBP1 with age.

Aging is associated with a prolonged UPR and unmitigated ER stress, which activates several apoptotic signaling cascades. UPR induces apoptosis via the PERK/eIF2α-mediated induction of CHOP and/or the IRE1-mediated activation of apoptosis signal-regulating kinase 1(ASK1)/JNK. Chhabra et al. found that cells overexpressing the miR-23a~27a~24-2 cluster upregulate components of the ER stress-mediated apoptosis pathway, i.e., C/EBP homologous protein (CHOP/DDIT3/GADD153) and TRIB3, an Akt inhibitor and ATF4. miR-23a~27a~24-2 also leads to a significant release of ER Ca2+ reserves into the cytoplasm and a concomitant increase in mitochondrial membrane permeability [151]. Another miRNA, miR-204, has also been implicated in apoptosis in response to oxidative and pharmacological ER stress. The overexpression of miR-204 attenuates the induction of several ER-responsive genes such as GRP78, GRP94, and CHOP and contributes to the phenotypes that are characteristic of senescent cells [152]. It will be interesting to determine whether age-associated elevation in oxidative stress controls miR-204 levels in order to regulate senescence.

Compared to aging studies, there have been many studies investigating ER stress–miRNA interactions in various age-associated metabolic and neurodegenerative diseases. Metabolic disorders such as obesity and insulin resistance have been associated with ER stress, especially since ER is the main organelle controlling protein and lipid metabolism and gluconeogenesis. Additionally, the expression of several miRNAs is differentially regulated under certain metabolic disorders [153]. For example, RIDD-mediated pre-miR-200 and pre-miR-34 degradation is involved in hepatic steatosis [144]. In turn, different miRNAs differentially regulate components of UPR branches. miR-30c, miR-708, and miR-143 play a role in the angiotensin II-mediated induction of ER stress and obesity in mice [154]. miR-34a, miR-122, and miR-30 were also determined to enhance the pathogenicity of nonalcoholic fatty liver disease (NAFLD) by regulating ER stress [155]. An ER stress–miRNA interaction has also been demonstrated to play a role in age-associated neurodegenerative diseases such as Alzheimer’s, Parkinson’s, amyloid lateral sclerosis (ALS), and Huntington’s. These pathologies involve the accumulation of misfolded protein aggregates, and UPR^ER^ plays a significant role in the development and progression of the disease [156]. For example, Aβ-induced ER stress inhibits PTEN expression by inducing miR-200c to protect neurons against Aβ toxicity [157]. miR-34a upregulation by IRE1 inhibition in SH-SY5Y (human neuroblastoma) cells has been reported to be protective against Aβ-mediated injury [158]. In addition, the inhibition of miR-34c and miR-34b expression in SH-SY5Y cells were demonstrated to aggravate PD pathogenesis by increasing α-synuclein [159]. miR-16-1 also contributes to PD development by inhibiting HSP70 levels and inducing α-synuclein accumulations in SH-SY5Y cells [160]. In contrast, miR-7 mediated ER stress suppression in these cells has been determined to be protective against PD [161]. Overall, miRNAs demonstrate huge potential in controlling age-associated diseases via regulating ER protein folding and UPR^ER^, and this avenue should be further explored.

### 4.3. Oxidative Stress Response

Increased oxidative stress (OS) is a prime hallmark of aging. With age, there is an increased production of reactive oxygen species, while the antioxidant enzymes such as catalases or superoxide dismutases decrease with age. Nuclear factor erythroid 2-related factor 2 (NRF2) is a transcription factor that regulates the cellular defense against toxic and oxidative insults through the expression of genes involved in oxidative stress response and drug detoxification [162]. NRF2 activity is tightly regulated through a complex transcriptional and post-translational network that enables it to orchestrate the cell’s response to various pathological stressors. Recently, miRNAs have been shown to be important players in controlling OS, aging, and cellular senescence across different organisms such as nematodes, flies, mice, and humans. NRF2 is also under the regulation of miRNAs [163]. For instance, in *C. elegans,* miR-228 forms a negative feedback loop with NRF2 homolog SKN-1 to regulate lifespan. *mir-228* mutant animals are long-lived and more stress resistant compared to wild-type animals. Importantly, the interaction between miR-228 and SKN-1 is also critical for the lifespan prolonging effects of dietary restriction (DR) [24]. Interestingly, the miRNA cluster miR-229,64,65,66, which is in the same family as miR-228, promotes longevity and stress resistance and forms an indirect positive feed-back loop with SKN-1. Both the miRNA cluster and SKN-1 promote each other’s expression and this relationship promotes longevity under DR, low-insulin signaling, and constitutive active SKN-1 [25]. NRF2 is also targeted by miR-144 in the cerebrovascular cells of aged rats. An age-associated increase in the expression of miR-144 and a concomitant decrease in NRF2 expression elevates oxidative stress with age. These effects were reversed by caloric restriction, which downregulated the miRNA expression and restored NRF2 levels, indicating that CR confers antioxidative effects through regulating the miR-144-NRF2 axis [164]. In a similar study performed using rat hepatocytes, age-related decreases in NRF2 levels were correlated with the levels of six miRNAs being significantly upregulated. Of these, miR-146a directly targets *Nrf2*, resulting in an age-associated decline in NRF2 levels in response to oxidative stress [165].

Several studies conducted in the heart, liver, or kidneys of different organisms elucidated the role and regulation of miRNAs in oxidative stress and antioxidant defense with age. For instance, Heid et al. observed that oxidative stress accumulation correlated with the differential expression of miR-29 in the heart of aged turquoise killifish *Nothobranchius furzeri*. Further, knocking down miR-29 in a zebrafish model resulted in morphological and cardiac alterations and impairments in oxygen-dependent pathways, suggesting that a miR-29 increase may prevent hypoxic cardiac damage with age [166]. Another study analyzed the miRNA expression profile of young and old rat kidneys and observed the upregulation of 18 miRNAs, with miR-335 and miR-34a exhibiting the most significant upregulation. Both of these miRNAs target genes in oxidative defense pathways, particularly superoxide dismutase 2 (SOD2) and thioredoxin reductase 2 (Txnrd2). The overexpression of miR-335 and miR-34a induced the premature senescence of young renal cells, while antisense miR-335 and miR-34a inhibited the senescence of old mesangial cells via the regulation of SOD2 and Txnrd2 levels, resulting in the suppression of reactive oxygen species (ROS) [167]. miRNAs also participate in the decline of oxidative defense mechanisms in the aging liver. The expression profiling of 367 miRNAs in livers from 4- to 33-month-old mice revealed age-associated increases in miR-669c, miR-709, miR-93, and miR-214. Further, these miRNAs target various classes of glutathione S-transferases and thus contribute to the age-related decline in oxidative defense mechanisms in the aging liver [168].

SIRT1 is a nicotinamide adenine dinucleotide- (NAD+−)-dependent deacetylase that regulates crucial cellular functions and is involved in combatting oxidative stress [169]. Crosstalk between ROS production and SIRT1 activity plays a crucial role in the regulation of the aging process [170]. A variety of miRNAs regulate SIRT1 expression [171]. For instance, miR-34a is known to target *SIRT1* and promote cellular senescence in different tissues [172,173]. Notably, miR-34a plays a crucial role in age-related macular degeneration (AMD) by regulating the oxidative stress resistance of aging retinal pigmented epithelium. The miRNA indirectly regulates the expression of the p66shc adaptor protein, a key protein that regulates cellular oxidative stress, by targeting SIRT1, a H3 deacetylase that regulates P66SHC expression. Recent studies indicate that in AMD, miR-34a is overexpressed, leading to SIRT1 decrease and a subsequent increase in P66SHC and oxidative stress [174]. miR-217 is also reported to target *SIRT1* and modulate its deacetylase activity, resulting in impaired angiogenesis in endothelial cells and a premature senescence phenotype [175].

Lastly, Argonaute proteins are under regulation by ROS. In a study to understand the molecular mechanism of RAS-induced premature senescence, ROS were found to inactivate protein tyrosine phosphatase 1B (PTP1B). This further dephosphorylated Argonaute protein AGO2, resulting in the inhibition of miRNA loading onto the Argonaute complex. This oxidative stress-mediated inhibition of miRNA function contributes to the induction of senescence [176]. Overall, miRNAs that can generate rapid and reversible responses to age-related oxidative stress play a vital role in modulating senescence and aging.

### 4.4. Autophagy

Autophagy is a homeostatic process by which a cell breaks down and eliminates misfolded proteins and dysfunctional organelles in a well-concerted pathway. It is a lysosomal-dependent mechanism that allows the orderly degradation and recycling of damaged cellular components within the cytoplasm. Like other processes, autophagy plays a central role in maintaining the proteome integrity of the cell. Autophagy comprises distinct stages, beginning with initiation/nucleation and followed by elongation, maturation, and lysosomal fusion, resulting in the degradation of autophagic cargo [177]. A complex containing VPS34, BECLIN1, ATG14, VPS15, and AMBRA1 controls nucleation by phosphorylating the membrane-associated phosphoinositol lipids (PI), resulting in the accumulation of phosphoinositol-3-phosphates (PI3Ps) on the cytoplasmic surface of organelle membranes [178]. This recruits lipid-binding proteins that serve as nucleation sites for autophagosome formation [178]. Next, elongation is mediated by two ubiquitinylation–conjugation systems involving ATG12-5-16 and ATG8/LC3-lipid conjugation systems. The former serves as an E3 ligase and facilitates the conjugation of ATG8 to phosphatidylethanolamine (PE), which further promotes membrane expansion and autophagic vesicle completion [179,180,181]. Later, autophago–lysosome fusion requires several SNARES, integral lysosomal proteins, and RAB proteins [182,183].

Recent studies point to the role of miRNAs in regulating autophagy during aging. In *C. elegans*, a secreted miRNA miR-83/miR-29 controls the age-related decrease in autophagy across different tissues. Activated by HSF-1, this miRNA can be transported to other tissues via extracellular vesicles where it disrupts a vital autophagy regulator, CUP-5, autonomously in intestine and non-autonomously in body wall muscles. Inhibiting miR-83 can upregulate macroautophagy in differential tissues, thus promoting proteostasis and longevity [184]. Additionally, loss-of-function mutations in miR-34, a highly conserved miRNA, significantly extend lifespan through the regulation of autophagy genes *atg-4.1, bec-1,* and *atg-9*. Similarly, mammalian miR-34a has been shown to directly bind and target ATG9 [26].

The relationship between miRNAs and different components of the autophagic process comes from the studies of various age-associated diseases. Recent reports indicate that miRNAs target autophagy genes that play an important role in the onset and development of age-related intervertebral disc degeneration (IDD) in humans [185]. For instance, miR-21 facilitates IDD progression by inhibiting *PTEN*-mediated autophagy and extracellular matrix degradation [186]. In some cases, miR-21 uses an IL-6 inflammatory response to blunt autophagy [187]. The autophagy-related genes (ATGs) 5 and 7 are also targeted by the miRNAs miR-153-3p, miR-202-5p, and miR-210 in human IDD samples [188,189,190]. Furthermore, miRNAs are also deemed responsible for regulating IDD by blocking upstream regulators of autophagy such as BECLIN1 (miR-129) and GALECTIN3 (miR-185) [191,192].

miRNAs are also known to play role in UVA- and UVB-induced photo-aging in human skin fibroblasts by regulating autophagy. For instance, the levels of miR-23 are upregulated during skin exposure to UV radiation, and inhibiting this miRNA using antagomirs stimulated the activation of autophagy and protected the human fibroblasts from UV-induced premature senescence. miR-23 targets the mRNA for the AMBRA1 protein, which is a positive regulator of the BECLIN1-VPS34 interaction to regulate autophagy [193]. miR-23 also plays a role in age-associated increases in oxidative stress in lens epithelial cells, one of the prime causes of eye cataracts with age. The knockdown of miR-23 is known to alleviate oxidative stress and apoptosis and increase autophagy in these cells by suppressing *SIRT1*, which contains binding sites for the miRNA in its 3′UTR [194].

miRNAs have also been implicated in regulating autophagy in age-associated neurodegenerative diseases. For example, the inhibition of miR-331-30 and miR-9-5p have been shown to prevent Alzheimer’s Disease progression by upregulating the autophagy required for the clearance of amyloid beta (Aβ) plaques. It has been demonstrated that miR-331-3p and miR-9-5p target the autophagy receptors Sequestosome 1 (*Sqstm1*) and Optineurin (*Optn*), respectively, and can also act as potential markers between early and late stages of the disease [195]. Another miRNA, miR-101, inhibits MAP kinase-dependent autophagy and, thus, its inhibition in AD is instrumental in alleviating pathogenesis by upregulating autophagy [196]. The role of miRNAs in regulating autophagic flux in other neurodegenerative diseases, particularly prion diseases, has been comprehensively reviewed elsewhere [197].

Autophagic clearance also plays an important role in the pathogenesis of age-related macular degeneration (AMD), which involves the detrimental aggregation of damaged proteins. Several miRNAs are dysregulated in AMD and many of them target various steps in the autophagic process (Table 2) [198]. Furthermore, miRNAs play a vital role in longevity-induced autophagy in cardiovascular tissues. Autophagy induction can be either protective or detrimental for the heart and, thus, the induction of autophagy by the longevity-promoting drug rapamycin is tightly regulated by post-transcriptional mechanisms such as cardiac–miRNA networks [199].

Finally, miRNAs also regulate mitophagy during age. Aging poses an irreversible risk for mitochondrial abnormalities, and thus, mitophagy initiation is induced to effectively recycle the damaged mitochondria. PTEN-induced putative kinase 1 (PINK1) acts as a stress sensor and effector inducer of mitophagy. A study by Tai et al. demonstrated that miR-34-5p inhibits mitophagy by directly targeting *Pink1* and causes the age-related downregulation of mitophagy in aged mice [200]. Overall, miRNAs form complex and bidirectional networks with autophagic processes and thus play an important role in the age-related collapse of proteostasis and organelle homeostasis.

## 5. Concluding Remarks

miRNA therapeutics, involving both miRNA mimics and inhibitors (AntimiR-122, AntimiR-21, AntimiR-103, AntimiR-17, Anti-miR-155, miR-29, miR-92, miR-16, and miR-34) are already in clinical trials for cancer and other diseases [201,202]. However, their potential to ameliorate diseases of aging have been less explored until now. This review has highlighted the role of several miRNAs that are highly conserved regulate lifespan across different organisms, and regulate proteome integrity with age. They target different components of the protein folding and degradation machinery and therefore carry a huge potential for future miRNA therapeutics for age-related protein misfolding diseases. Others target the key components of the intracellular stress response pathways that detect the loss of homeostasis and restore it. Certain miRNAs, such as miR-34, *let-7*, and miR-125, are highly conserved and play a role in lifespan regulation in invertebrate models, and are differentially expressed in several age-associated pathologies [14,26,29,33,96,173]. However, a mechanistic understanding of their role in age-associated proteostasis and stress defenses is superficially characterized. One exception is the highly studied miR-34, which not only regulates lifespan in nematodes and flies, but also plays a role in brain, renal, and liver aging in mice and humans. miR-34 is a target of several stress response transcription factors such as HSF1, FOXO3A, and IRE1-XBP1-mediated RIDD and, in turn, regulates the expression of chaperones and other key components of these stress response pathways [26,33,96,120,146,158,159,167,173,200]. Thus, this suggests a huge therapeutic potential in manipulating the levels of miR-34 in treatments for diseases that involve the accumulation of misfolded proteins. However, it is important to understand whether its deregulation in aging is the cause or just a marker of age-associated diseases. Additionally, a meticulous understanding of its temporal regulation in an organ specific manner is required in order to maximize the precision and specificity of its curative potential.

## Figures and Tables

**Figure 1 ncrna-09-00026-f001:**
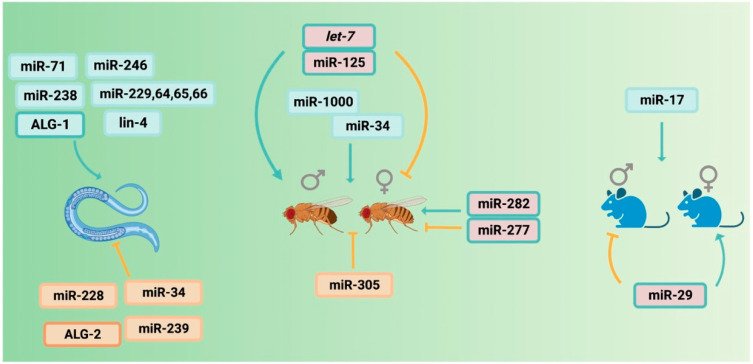
MiRNAs regulating lifespan in nematodes, flies, and mice. Schematic represents miRNAs that regulate lifespan (either positively or negatively) in *C. elegans*, *Drosophila melanogaster*, and *Mus muluscus.* In flies and mice, miRNAs regulating lifespan in a sex-specific manner are shown via different arrows (blue: promote lifespan, orange: inhibit lifespan).

**Figure 2 ncrna-09-00026-f002:**
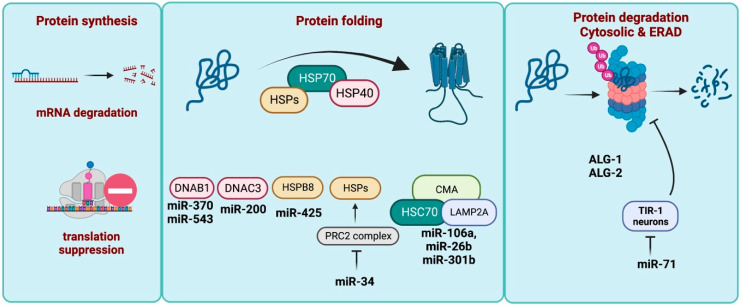
miRNAs regulating different components of proteostasis machinery. miRNAs regulating different components of protein folding and degradation machinery are shown. Different heat shock proteins (HSPs) including small HSPs (HSP40s or DNAJ proteins in pink) are shown along the miRNA targeting them.

**Table 2 ncrna-09-00026-t002:** miRNAs involved in regulation of autophagic processes in various age-associated diseases.

Lifespan/Age-Associated Disease	miRNAs	Autophagic Processes	References
Lifespan	miR-83 miR-34	CUP-5 ATG-4.1, BEC-1, AND ATG-9	Zhou, Y. et al., 2019 [184] Yang, J. et al., 2013 [26]
Intervertebral disc degeneration (IDD)	miR-21 miR-153-3p, miR-202-5p, and miR-210 miR-129, miR-185	PTEN ATG5, ATG7 BECN-1, GALECTIN3	Wang, W. J. et al., 2018 [186] Wang, C. et al., 2017 [188] Chen, J. et al., 2020 [189,190] Wang, X. B, 2019 [190] Zhao, K. et al., 2017 [191] Yun, Z. et al., 2020 [192]
Photo-aging	miR-23	AMBRA1	Zhang, J. et al., 2016 [193]
Cataract	miR-23	SIRT1	Zhou, W. et.al, 2019 [194]
Alzheimer’s	miR-331-30, miR-9-5p miR-101	SQSTM1, OPTN MAPK1	Chen, M. L. et al., 2021 [195] Li, Q. et al., 2019 [196]
Age-related macular degeneration	miR-9, miR-124, miR-17, miR-29-3p, miR-129-3p, let-7, miR-335, miR-378, miR-26b, miR-20a, miR-21, miR-205, miR-34a, miR-146, miR-155, miR-132	Induction	Hyttinen, J. M. T., et.al., 2021 [198]
	miR-146b, miR-205, miR-342-3p, miR34a, miR-106a, miR-17, miR-20a, miR-124 and miR-361	Nucleation	
	miR-21, miR-200c, miR-361, miR-20a, miR-24-3p, miR-129-3p, miR-204, miR-125b and miR-206	Elongation and completion	
	miR-17, miR-21, miR-150, miR-184, miR124 and miR-26b	Fusion and degradation	
Age-associated mitophagy	miR-34-5p	PINK1	Tai, Y. et al., 2021 [200]

## Data Availability

Not applicable.

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
