# Peer review of "MicroRNAs in Age-Related Proteostasis and Stress Responses"

_ncrna, 2023, doi:10.3390/ncrna9020026_

Round 1

Reviewer 1 Report

The review submitted by Matai and Slack focuses on the role of miRNAs in proteostasis and stress responses, two processes with fundamental impact on cellular and organismal aging. Given the growing evidence of miRNAs being causally related to many aspects of aging, this review is timely and provides an extensive overview about the recent developments in the field.

While the manuscript is of good quality, a more succinct organization, especially of chapter 4 and its sub-parts, would aid readability. My suggestion would be to structure chapter 4 according to the most important miRNAs, instead of separating different stresses. Thereby, the partly redundant explanation of miRNA functions in the four mentioned conditions (heat shock response, ER stress response, oxidative stress response and autophagy) would be avoided. Other minor drawbacks are the multiple introduction of abbreviations and their inconsistent usage as well as the sometimes incorrect and changing nomenclature of genes and proteins. Moreover, throughout the text, individual miRNAs are written sometimes in italics and sometimes not. Below, I am listing parts of the manuscript that need correction/improvement (page X, line Y abbreviated as pX, lY):

-       p2,l65; p4, l181; p7, l280: Please unify abbreviations for insulin/ insulin-like growth factor signaling.

-       p2, l72: The abbreviation of “loss of function” (lof) is not consistently used and thus superfluous.

-       p2, l89: “…and it is generally sex specific” Does “it” mean “this effect”?

-       p3, l107/108: Abbreviation “KD” is not introduced.

-       p3, l114: Please correct “Map-kinase”.

-       p3, l114/115: Please correct “n humans”

-       p3, l129: Please correct “Sirtuin1”

-       p3, l129-154: Please correct gene nomenclature (sirt1, bcl2, bace1 and pten are neither mouse nor human genes).

-       p3, l140: Space is missing (“miR-431are”)

-       p3, l149: Please correct “Levels…increases with age and is…”

-       p4, l155-157: Which miRNA is meant miR-34 (l155) or miR-34a (l157)?

-       p4, l169: Please correct “Mus muluscus” and have the species names in italics. Please also provide a proper figure legend and not only the title.

-       p4, l185/186: Do you mean “translational programming” or “reprogramming”?

-       p5,l191-193: Please clarify that “HSP40 chaperone” belong to the “small Hsps” class. It would also make more sense to use plural (as it is a family of proteins). Also, DnaJ-proteins as an alternative name (instead of “J-proteins”) would be more precise and would help to understand the following text mentioning “DNAJB1”.

-       p5, l214: A bit more background than “H3K27me3 methyltransferase” about the histone H3 lysine 27 trimethylating function of PRC2 would be beneficial for readers not familiar with epigenetics.

-       p5, l233: “on polyubiquitylation”; Was “upon polyubiquitylation” meant?

-       p5, l234: “on animals”; Was “in animals” meant?

-       p5, l235: “with alg-2 RNAi but not alg-1”; Please add “RNAi”.

-       p5, l235: The term “gerontomiR” is not explained.

-       p5, l239: TIR1 = toll/interleukin1 receptor 1

-       p6, l240: “which is important for the effect of food perception proteosome” Please add “on” after “perception”.

-        p6, l243/244: Please superscript “2+” in Ca2+.

-       p6, l253: “Bip” belongs to the HSP70 family, but to avoid confusion I would suggest to give the full name (heat shock 70 kDa protein 5 (HSPA5)).

-        p6, l254: Also Fig. 2 needs a proper legend and not only a title.

-       p7, l286: Abbreviation “ncRNA” is superfluous.

-       p7, l320: Please change “bonafide” to “bona fide”.

-       p8, l339: Please change “the core protein required for miRNA gene silencing” to “the core protein required for miRNA-dependent gene silencing”.

-       p8, l344: Please provide more information about “H9C2” cells.

-       p8/9, Table 1: Please correct the typo “Schereiner”. The table does not give too much information and could be more compacted, e.g. by not having only on miRNA per line.

-       p9, l353: Please superscript “ER” in “UPRER” here and throughout the manuscript.

-       p9, l355/356: PERK stands for “Protein kinase R (PKR)-like Endoplasmic Reticulum Kinase”

-       p9, l365: “spliced XBP1 functions as a transcription factor”. Please correct to, “the translation product of the spliced XBP1 mRNA is a transcription factor”.

-       p10, l394-396: Please correct “mRNA” to “miRNA” (as RIDD targets miRNAs and not mRNAs).

-       p11, l438: First time that abbreviations for Alzheimer’s and Parkinson’s disease are introduced, albeit these diseases have been mentioned several times above.

-        p11, l443: Please provide more information about the SH-SY5Y cells.

-       p11, l482/483: “Nothobranchius furzeri (Nfu)”; Please write in italics and remove “(Nfu)”.

-       p12, l505-507: “The miRNA indirectly regulates the expression of p66shc adaptor protein, a key protein 505 that regulates cellular oxidative stress by targeting SIRT1, a H3 deacetylase that regulates 506 P66SHC expression.” This sentence is hard to read. At least ad a comma after “stress”.

-       p13, l551: Please correct “In Some cases”.

-       p14, Table 2: The third column “Autophagy gene target” should be renamed as also autophagic processes are listed.

Author Response

We thank the reviewers for the positive comments and helpful critique.

Reviewer1

Comments and Suggestions for Authors

The review submitted by Matai and Slack focuses on the role of miRNAs in proteostasis and stress responses, two processes with fundamental impact on cellular and organismal aging. Given the growing evidence of miRNAs being causally related to many aspects of aging, this review is timely and provides an extensive overview about the recent developments in the field.

While the manuscript is of good quality, a more succinct organization, especially of chapter 4 and its sub-parts, would aid readability. My suggestion would be to structure chapter 4 according to the most important miRNAs, instead of separating different stresses. Thereby, the partly redundant explanation of miRNA functions in the four mentioned conditions (heat shock response, ER stress response, oxidative stress response and autophagy) would be avoided. Other minor drawbacks are the multiple introductions of abbreviations and their inconsistent usage as well as the sometimes incorrect and changing nomenclature of genes and proteins. Moreover, throughout the text, individual miRNAs are written sometimes in italics and sometimes not. Below, I am listing parts of the manuscript that need correction/improvement (page X, line Y abbreviated as pX, lY):

We appreciate reviewer’s suggestion on restructuring part 4 of the manuscript. However, there aren’t many miRNAs that play a role in several stress responses, therefore reorganizing it according to the most mentioned miRNAs will be a little difficult. However, we have added an introductory paragraph to this section mentioning the stresses getting discussed in this section and highlighting the miRNAs that play a role in multiple stresses. 

-       p2,l65; p4, l181; p7, l280: Please unify abbreviations for insulin/ insulin-like growth factor signaling. We corrected this.

-       p2, l72: The abbreviation of “loss of function” (lof) is not consistently used and thus superfluous. We corrected this.

  •       p2, l89: “…and it is generally sex specific” Does “it” mean “this effect” We meant and added ‘this effect’.

-       p3, l107/108: Abbreviation “KD” is not introduced. We introduced this abbreviation.  

-       p3, l114: Please correct “Map-kinase”. We corrected this.

-       p3, l114/115: Please correct “n humans” We corrected this.

-       p3, l129: Please correct “Sirtuin1” We corrected this.

-       p3, l129-154: Please correct gene nomenclature (sirt1, bcl2, bace1 and pten are neither mouse nor human genes). We corrected this.

-       p3, l140: Space is missing (“miR-431are”) We corrected this.

-       p3, l149: Please correct “Levels…increases with age and is…” We are unable to understand the reviewer’s comment here.

-       p4, l155-157: Which miRNA is meant miR-34 (l155) or miR-34a (l157)? We aren’t sure about this therefore modified this sentence.

-       p4, l169: Please correct “Mus muluscus” and have the species names in italics. Please also provide a proper figure legend and not only the title. We corrected this.

-       p4, l185/186: Do you mean “translational programming” or “reprogramming”? It is Reprogramming. We corrected this.

-       p5,l191-193: Please clarify that “HSP40 chaperone” belong to the “small Hsps” class. It would also make more sense to use plural (as it is a family of proteins). Also, DnaJ-proteins as an alternative name (instead of “J-proteins”) would be more precise and would help to understand the following text mentioning “DNAJB1”. We added this information according to the reviewer’s suggestion.

-       p5, l214: A bit more background than “H3K27me3 methyltransferase” about the histone H3 lysine 27 trimethylating function of PRC2 would be beneficial for readers not familiar with epigenetics. We added this information according to the reviewer’s suggestion.

-       p5, l233: “on polyubiquitylation”; Was “upon polyubiquitylation” meant? We corrected this to ‘upon polyubiquitylation’.

-       p5, l234: “on animals”; Was “in animals” meant? We corrected this.

-       p5, l235: “with alg-2 RNAi but not alg-1”; Please add “RNAi”. Added.

-       p5, l235: The term “gerontomiR” is not explained. We added the explanation.

-       p5, l239: TIR1 = toll/interleukin1 receptor 1 We corrected this.

-       p6, l240: “which is important for the effect of food perception proteosome” Please add “on” after “perception”. We corrected this.

-        p6, l243/244: Please superscript “2+” in Ca2+. We corrected this.

-       p6, l253: “Bip” belongs to the HSP70 family, but to avoid confusion I would suggest to give the full name (heat shock 70 kDa protein 5 (HSPA5)). We added this information according to the reviewer’s suggestion.

-        p6, l254: Also Fig. 2 needs a proper legend and not only a title. We added this information according to the reviewer’s suggestion.

-       p7, l286: Abbreviation “ncRNA” is superfluous. We corrected this.

-       p7, l320: Please change “bonafide” to “bona fide”. We corrected this.

-       p8, l339: Please change “the core protein required for miRNA gene silencing” to “the core protein required for miRNA-dependent gene silencing”. We corrected this.

-       p8, l344: Please provide more information about “H9C2” cells. We added this information according to the reviewer’s suggestion.

-       p8/9, Table 1: Please correct the typo “Schereiner”. The table does not give too much information and could be more compacted, e.g. by not having only on miRNA per line. We modified the table according to reviewer’s suggestion.

-       p9, l353: Please superscript “ER” in “UPRER” here and throughout the manuscript. We corrected this.

-       p9, l355/356: PERK stands for “Protein kinase R (PKR)-like Endoplasmic Reticulum Kinase” We corrected this.

-       p9, l365: “spliced XBP1 functions as a transcription factor”. Please correct to, “the translation product of the spliced XBP1 mRNA is a transcription factor”. We corrected this.

-       p10, l394-396: Please correct “mRNA” to “miRNA” (as RIDD targets miRNAs and not mRNAs). We did not understand this comment. RIDD targets both miRNA and mRNAs.

-       p11, l438: First time that abbreviations for Alzheimer’s and Parkinson’s disease are introduced, albeit these diseases have been mentioned several times above.  We corrected this.

-        p11, l443: Please provide more information about the SH-SY5Y cells. We added this information according to the reviewer’s suggestion.

-       p11, l482/483: “Nothobranchius furzeri (Nfu)”; Please write in italics and remove “(Nfu)”. We corrected this.

-       p12, l505-507: “The miRNA indirectly regulates the expression of p66shc adaptor protein, a key protein 505 that regulates cellular oxidative stress by targeting SIRT1, a H3 deacetylase that regulates 506 P66SHC expression.” This sentence is hard to read. At least ad a comma after “stress”. We modified this

-       p13, l551: Please correct “In Some cases”. We corrected this.

-       p14, Table 2: The third column “Autophagy gene target” should be renamed as also autophagic processes are listed. We corrected this.

Reviewer 2 Report

Review entitled” microRNAs in age-related proteostasis and stress responses” is well written and would be very helpful for the researchers studying role of microRNAs in ageing. However, there are few shortcomings that readers may have

Comments

Ø In the introduction section, the authors should briefly describe about the biogenesis of microRNA

Ø In line 60, the authors have mentioned that microRNA serve as potential biomarkers for aging. Is there any specific miR that is being widely as a biomarker for aging

Ø In line 114, correct “n” as “In”

Ø Is there any miRNA that is under clinical trials for improving age related disorders

Ø In line 353 the authors should be clear whether it is  “UPRER”  or   “UPR or ER” as in previous lines the author has mentioned unfolded protein response or ER stress response

Ø In the conclusion remarks the authors have mentioned that miRNA therapeutics are already making their way to clinical trials. It would be better if the authors will provide the list of miRs that are in clinical trials.

Author Response

We thank the reviewers for the positive comments and helpful critique.

Reviewer2

Comments and Suggestions for Authors

Review entitled” microRNAs in age-related proteostasis and stress responses” is well written and would be very helpful for the researchers studying role of microRNAs in ageing. However, there are few shortcomings that readers may have

Comments

Ø In the introduction section, the authors should briefly describe about the biogenesis of microRNA. We have added this information according to the reviewer’s suggestion.

Ø In line 60, the authors have mentioned that microRNA serve as potential biomarkers for aging. Is there any specific miR that is being widely as a biomarker for aging? We have added the reference that reviews this information. Briefly, miRNAs (like miR-34) that change their expression with age ubiquitously or in a specific tissue can act as potential aging biomarkers.

Ø In line 114, correct “n” as “In” We corrected this.

Ø Is there any miRNA that is under clinical trials for improving age related disorders. We have now added this information briefly mentioning miRNAs that are currently under clinical trials.

Ø In line 353 the authors should be clear whether it is  “UPRER”  or   “UPR or ER” as in previous lines the author has mentioned unfolded protein response or ER stress response. We have corrected UPRER to UPRER to avoid this information. UPRER (UPRER) and ER stress response mean the same thing.  

Ø In the conclusion remarks the authors have mentioned that miRNA therapeutics are already making their way to clinical trials. It would be better if the authors will provide the list of miRs that are in clinical trials. We have now added this information briefly mentioning miRNAs that are currently under clinical trials.

Reviewer 3 Report

The manuscript is clearly and comprehensibly written. Each chapter begins with an introduction which makes it easier to understand the authors' intentions. Overviews describing the presented topic are available in the literature databases. However, extending the age-related proteostasis theme to include different stress responses brings new insights to the issue. 

However, the editorial side of the text needs to be improved:

line114; 140;217;287.

Author Response

Reviewer3

The manuscript is clearly and comprehensibly written. Each chapter begins with an introduction which makes it easier to understand the authors' intentions. Overviews describing the presented topic are available in the literature databases. However, extending the age-related proteostasis theme to include different stress responses brings new insights to the issue. 

However, the editorial side of the text needs to be improved:

line114; 140;217;287  Thankyou for your compliments. We corrected these lines now to the best of our understanding.